materials science/inorganic chemistry

oxides, TFT-LCD, Raman spectroscopy, physico-chemical properties

**Author for correspondence:**
Jiedong Cui
e-mail: jdcui04@163.com

# Influence of the slight adjustment of oxides on the structural and physico-chemical properties of thin film transistor-liquid crystal display substrate glass

Jiedong Cui[1,2], Xin Cao[1,2], Lifen Shi[1,2], Zhaojin Zhong[1,2], Pingping Wang[1,2] and Liyun Ma[1,2]

[1]State Key Laboratory of Advanced Technology for Float Glass, No. 751 Donghai Road, Bengbu, Anhui 233018, People's Republic of China
[2]Bengbu Design & Research Institute for Glass Industry Co., Ltd. No. 1047 Tushan Road, Bengbu, Anhui 233018, People's Republic of China

JC, 0000-0003-2518-0530

By the slight adjustment of oxides constituting thin film transistor-liquid crystal display (TFT-LCD) substrate glass, including equal mole fraction substitution of $Al_2O_3$, $GeO_2$, $B_2O_3$, $P_2O_5$ and $ZrO_2$ for $SiO_2$, as well as the substitution of CaO for SrO with the total contents unchanged, the structural and physico-chemical properties of the glass was investigated by Raman spectroscopy and other measurements. The results showed that the short-range disorder brought by the substitution of $GeO_2$, $B_2O_3$ and $P_2O_5$ for $SiO_2$ could weaken the stability and compactness of the glass network, and the physico-chemical properties deteriorated, while the process of glass melting would become easier accordingly. The short-range disorder by the substitution of $ZrO_2$ for $SiO_2$ with 1% mole fraction showed a little difference with other samples. Finally, the substitution of modified cations, such as CaO and SrO, showed a smaller variation compared with the substitution of network formers. On the condition of 1% mole fraction substitution of oxides investigated, the variation of samples showed a reasonable change and the performance was basically all satisfied for the use of TFT-LCD substrate.

# 1. Introduction

Thin film transistor-liquid crystal display (TFT-LCD) substrate glass is a kind of substrate for electronic display [1,2]. In order to meet the demand of the process of LCD devices, the glass needs to be alkali-free. The international representative products such as Corning Eagle XG™ and Asahi AN100™ mainly contain some conventional oxides, such as $SiO_2$, $Al_2O_3$, $B_2O_3$, MgO, CaO and SrO [3,4]. The influence of some unconventional oxides on the structural and physico-chemical properties of alkali-free glass deserve further study for the improvement of the glass industry. For example, some glass network formers, such as $GeO_2$ and $P_2O_5$, and the modified cations $ZrO_2$, the substitution of them for $SiO_2$ could play an important role in shaping the glass network and improving the performance of glass. Herein, we studied the influence of the slight adjustment of oxides mentioned above. To investigate the structural and physico-chemical properties of the glass, Raman spectroscopy, as well as some physico-chemical measurements were used to evaluate the effects of the slight adjustment of oxides.

# 2. Experiment

## 2.1. The chemical composition of glass

Seven kinds of the chemical compositions of alkali-free glass for TFT-LCD substrate are summarized in table 1, wherein sample 1 is the reference sample for the other six samples, which are acquired by the slight adjustment of oxides from sample 1. The details are as follows: sample 2 is acquired by 1% mole fraction substitution of $Al_2O_3$ for $SiO_2$ with equal total amount compared with sample 1, and the other samples are acquired in the same way as sample 2, including 1% mole fraction substitution of $GeO_2$ for $SiO_2$ for sample 3, 1% mole fraction substitution of $B_2O_3$ for $SiO_2$ for sample 4, 1% mole fraction substitution of $P_2O_5$ for $SiO_2$ for sample 5, 1% mole fraction substitution of SrO and CaO for sample 6 and 1% mole fraction substitution of $ZrO_2$ for $SiO_2$ for sample 7. By the slight adjustment of the oxides, the structural along with the physico-chemical properties were investigated.

## 2.2. Glass preparation

According to the chemical compositions shown in table 1, the weighted and evenly mixed glass batch was heated in a platinum crucible in a Silicon Molybdenum Furnace with the following melting and annealing procedure: (i) heating from room temperature up to 1000°C with a heating rate of 4°C min$^{-1}$; (ii) heating from 1000°C up to 1640°C with a heating rate of 2°C min$^{-1}$; (iii) dwelling in 1640°C for 2 h for fining of glass liquid; (iv) pouring the glass liquid onto a copper plate for glass formation; and (v) annealing at 650°C for 1 h in a muffle furnace and then cooling down to room temperature. The prepared glass samples were cut and ground for the following measurements.

## 2.3. Measurements

Samples without visible bubbles were selected for the following measurements: (i) Raman spectroscopy test with the wavenumber from 400 cm$^{-1}$ to 1600 cm$^{-1}$, Renishaw, Qontor Invia Raman spectrometer; (ii) density, Sartorius Stedim, QUINTIX224-1CN densitometer; (iii) elastic modulus and shear modulus test, GindoSonic, MK7 emodumeter; (iv) transmittance, Hitachi, U-4100 uv/vis spectrophotometer; (v) refractive index, Atago, NAR-4T abbe refractometer; (vi) resistivity at 25°C and 250°C, Bei Guang Jing Yi Instrument, BEST-212 resistometer; (vii) resistivity at high temperature from 1300°C to 1600°C, Xuhui Xin Rui, GHTR-1600 high temperature resistometer; (viii) thermal expansion, Linseis, L75 VS1000 dilatometer; (ix) strain point and annealing point; Orton, ANS-1000 LW strain point and annealing point tester; (x) softening point, Orton, SP 1000 LW softening point tester; and (ix) high temperature viscosity, Orton, RSV 1600 rotary viscometer.

# 3. Results and discussion

## 3.1. Structure of glass

The effects of the slight adjustment of oxides on the structure of the glass was studied by Raman spectroscopy. According to the theory of irregular network of glass, the average non-bridging oxygens

**Table 1.** Chemical compositions of TFT-LCD substrate glass.

| sample | chemistry composition (mol%) | | | | | | | | | | remarks |
| | $SiO_2$ | $GeO_2$ | $Al_2O_3$ | $B_2O_3$ | $P_2O_5$ | CaO | MgO | SrO | $ZrO_2$ | SnO | |
|---|---|---|---|---|---|---|---|---|---|---|---|
| 1 | 66.00 | 0.00 | 10.70 | 7.60 | 0.00 | 5.00 | 5.80 | 4.80 | 0.00 | 0.10 | reference |
| 2 | 65.00 | 0.00 | 11.70 | 7.60 | 0.00 | 5.00 | 5.80 | 4.80 | 0.00 | 0.10 | $Al_2O_3 \leftrightarrow SiO_2$ |
| 3 | 65.00 | 1.00 | 10.70 | 7.60 | 0.00 | 5.00 | 5.80 | 4.80 | 0.00 | 0.10 | $GeO_2 \leftrightarrow SiO_2$ |
| 4 | 65.00 | 0.00 | 10.70 | 8.60 | 0.00 | 5.00 | 5.80 | 4.80 | 0.00 | 0.10 | $B_2O_3 \leftrightarrow SiO_2$ |
| 5 | 65.00 | 0.00 | 10.70 | 7.60 | 1.00 | 5.00 | 5.80 | 4.80 | 0.00 | 0.10 | $P_2O_5 \leftrightarrow SiO_2$ |
| 6 | 66.00 | 0.00 | 10.70 | 7.60 | 0.00 | 6.00 | 5.80 | 3.80 | 0.00 | 0.10 | $CaO \leftrightarrow SrO$ |
| 7 | 65.00 | 0.00 | 10.70 | 7.60 | 0.00 | 5.00 | 5.80 | 4.80 | 1.00 | 0.10 | $ZrO_2 \leftrightarrow SiO_2$ |

**Table 2.** R, Z, Y, X of TFT-LCD substrate glass.

| sample | $RO/Al_2O_3$ | $[RO-Al_2O_3]/B_2O_3$ | O | $SiO_2 + GeO_2 + Al_2O_3 + B_2O_3 + P_2O_5$ | R | Z | Y | X |
|---|---|---|---|---|---|---|---|---|
| 1 | 1.46 | 0.65 | 202.60 | 102.60 | 1.97 | 3.85 | 3.75 | 0.10 |
| 2 | 1.33 | 0.51 | 203.60 | 103.60 | 1.97 | 3.85 | 3.78 | 0.08 |
| 3 | 1.46 | 0.64 | 202.60 | 102.60 | 1.97 | 3.85 | 3.75 | 0.10 |
| 4 | 1.46 | 0.57 | 203.60 | 103.60 | 1.97 | 3.83 | 3.74 | 0.10 |
| 5 | 1.46 | 0.64 | 205.60 | 103.60 | 1.98 | 3.85 | 3.74 | 0.12 |
| 6 | 1.46 | 0.64 | 202.60 | 102.60 | 1.97 | 3.85 | 3.75 | 0.10 |
| 7 | 1.46 | 0.64 | 202.60 | 101.60 | 1.99 | 3.85 | 3.71 | 0.14 |

$O_{nb}$ in oxygen polyhedron is expressed by $X$, the average bridging oxygens $O_b$ is expressed by $Y$, and $Z$ represents the average coordination number of network formers [5–7]. As a result of $RO/Al_2O_3 > 1$, $Al_2O_3$ will be treated as network former. So the network formers are as follows: $SiO_2$, $GeO_2$, $Al_2O_3$, $B_2O_3$ and $P_2O_5$, while MgO, CaO, SrO and $ZrO_2$ are treated as modified cations. $Al^{3+}$ shows the priority to be coordinated by oxygen ions to form an [$AlO_4$] tetrahedron compared with $B^{3+}$ [8]. As shown in table 1, $RO/Al_2O_3 > 1$ and $(RO-Al_2O_3)/B_2O_3 < 1$, so $Al^{3+}$ will mainly exist in the form of an [$AlO_4$] tetrahedron, and $B^{3+}$ will enter the network with more [$BO_3$] triangle and less [$BO_4$] tetrahedron. Table 2 shows the average coordination number $Z$ of seven samples, wherein $R$ represents the ratio of oxygen ions and network former ions, $X = 2R - Z$ and $Y = 2Z - 2R$ [5–7].

As a result of the similar components for all samples, the architecture of the glass network basically did not show too many differences, which are mainly dominated by the closer network formers such as $SiO_2$, $B_2O_3$ and $Al_2O_3$, as well as the same total amount of modified cations of CaO, MgO and SrO. As shown in figure 1, Raman peaks for all seven samples were mainly located at three frequencies with the closer intensity, indicating that the entity and compactness of the glass network changed a little with the substitution of oxides for $SiO_2$, proved by the closer $Y$ and $X$ values shown in table 2. The highest peak intensity was distributed in low frequency at around 480 cm$^{-1}$ assigning to the bending vibration of Si–$O_b$ in Si–$O_b$–Si bonds [9]. The second highest peak intensity was in the region of 850–1250 cm$^{-1}$ relating to the stretching vibration of bridging oxygens Si–$O_b$ and the stretching vibration of non-bridging oxygens Si–$O_{nb}$ [10,11]. The third highest peak intensity was in the middle frequency at around 803 cm$^{-1}$ relating to the symmetric stretching vibration of Si–$O_b$–Si between [$SiO_4$] tetrahedrons [9,12]. Some weaker intensity peaks distributed at around 710 cm$^{-1}$, 1310 cm$^{-1}$ and 1440 cm$^{-1}$ could attribute to the vibration of the B–O bond, wherein a peak at 710 cm$^{-1}$ is assigned to B–O–B bending vibration in the [$BO_3$] triangle [13], and peaks located at around 1310 cm$^{-1}$ and around 1440 cm$^{-1}$ are assigned to B–$O_b$ stretching vibration and B–$O_{nb}$ stretching vibration in the [$BO_3$] triangle, respectively [14].

Peaks at around 480 cm$^{-1}$ and 803 cm$^{-1}$ show very similar frequency and intensity for all seven samples, owing to the closer bridging oxygens, but there were still a few differences with sample

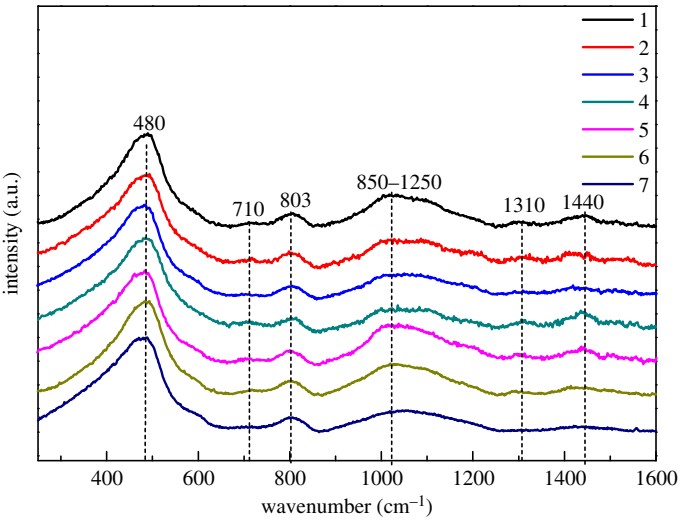

**Figure 1.** Raman spectra of TFT-LCD substrate glass.

1. Peaks located at around 480 cm$^{-1}$ tended to broaden and to shift to lower frequency for samples 2, 3, 4, 5 and 7, indicating the increase of the short-range disorder degree of network with the substitution of oxides for SiO$_2$, while sample 6 with the substitution of modified cations of SrO and CaO showed nearly the same as sample 1. Sample 5 with substitution of P$_2$O$_5$ for SiO$_2$ showed smaller intensity at 803 cm$^{-1}$, mainly owing to the P=O double bonds.

Peaks at around 710 cm$^{-1}$, 1310 cm$^{-1}$ and 1440 cm$^{-1}$ referring to the B–O stretching vibration were similar for samples 1, 2 and 3. Sample 4 with highest B$_2$O$_3$ content showed higher peak intensity at around 710 cm$^{-1}$ and 1440 cm$^{-1}$, in accordance with the increasing [BO$_3$] triangles brought by B$_2$O$_3$. For sample 6, the intensity of peaks weakened with the decrease of SrO. With the increase of the radius of modified cations, the restrainment of oxygen ions by the cations weakened. O$^{2-}$ can coordinate B$^{3+}$ to form a [BO$_4$] tetrahedron and thus lowered the [BO$_3$] triangle, causing the weakening of B–O stretching vibration.

Figure 2*a* shows Gaussian fit of Raman peaks in the region of 850–1250 cm$^{-1}$. Generally, peaks in this region refer to stretching vibration of bridging oxygens Si–O$_b$ and non-bridging oxygens Si–O$_{nb}$ in different Q$^n$ structure units, wherein n represents the number of bridging oxygens in the [SiO$_4$] tetrahedron [15]. As the average bridging oxygens in oxygen polyhedron approaching 3.75 as a roughly estimated Y value shown in table 2, the [SiO$_4$] tetrahedrons are mainly comprised of the Q$^3$ and Q$^4$ units, which refer to the [SiO$_4$] tetrahedron with 3 and 4 bridging oxygens, respectively.

For samples 1–7, there were three main fit peaks in this region. The first peak with the highest intensity was at a frequency of around 1030 cm$^{-1}$, corresponding to stretching vibration of bridging oxygens Si–O$_b$. Some researchers assumed that this peak should be owing to the vibration of bridging oxygens except Q$^4$ unit [16,17]. As Q$^1$ and Q$^2$ units were not obviously found in this study, the peak located at around 1030 cm$^{-1}$ should be derived from the Q$^3$ unit. The second highest peak intensity located at around 1170 cm$^{-1}$ corresponded to stretching vibration of bridging oxygens Si–O$_b$ in the Q$^4$ unit [16,17]. The weakest peak was located at around 1110 cm$^{-1}$, corresponding to stretching vibration of non-bridging oxygens Si–O$_{nb}$ in the Q$^3$ unit. As shown in figure 2*a*, peaks at around 1030 cm$^{-1}$ and 1170 cm$^{-1}$ accounted for most of the region, indicating that the vibration of bridging oxygens Si–O$_b$ dominates in alkali-free glasses, in accordance with the higher Y value roughly estimated in table 2.

The increase of disorder degree of the network can reduce the most probable angle and increase the bond length of Si–O$_b$–Si bond, which could make the intensity of chemical bond weaken, and thus decrease the frequency of stretching vibration of bridging oxygens Si–O$_b$ [18,19]. As shown in figure 2*b*, compared with reference sample 1, the intensity of peaks located at around 1030 cm$^{-1}$ became weaker. Samples 1 and 6 presented higher intensity than other samples, owing to their most SiO$_2$ content. Samples 2, 3, 4, 5 and 7 contained the same SiO$_2$ content, and the intensity was then dominated by the Y value. Samples 4, 5 and 7 showed smaller intensity owing to the lower Y value than samples 2 and 3. So on condition of the same SiO$_2$ content, the intensity of peaks located at around 1030 cm$^{-1}$ basically corresponded to the Y value as shown in figure 2*b*, and the tendency to lower intensity with the substitution for SiO$_2$ indicated the weakening of vibration of bridging oxygens.

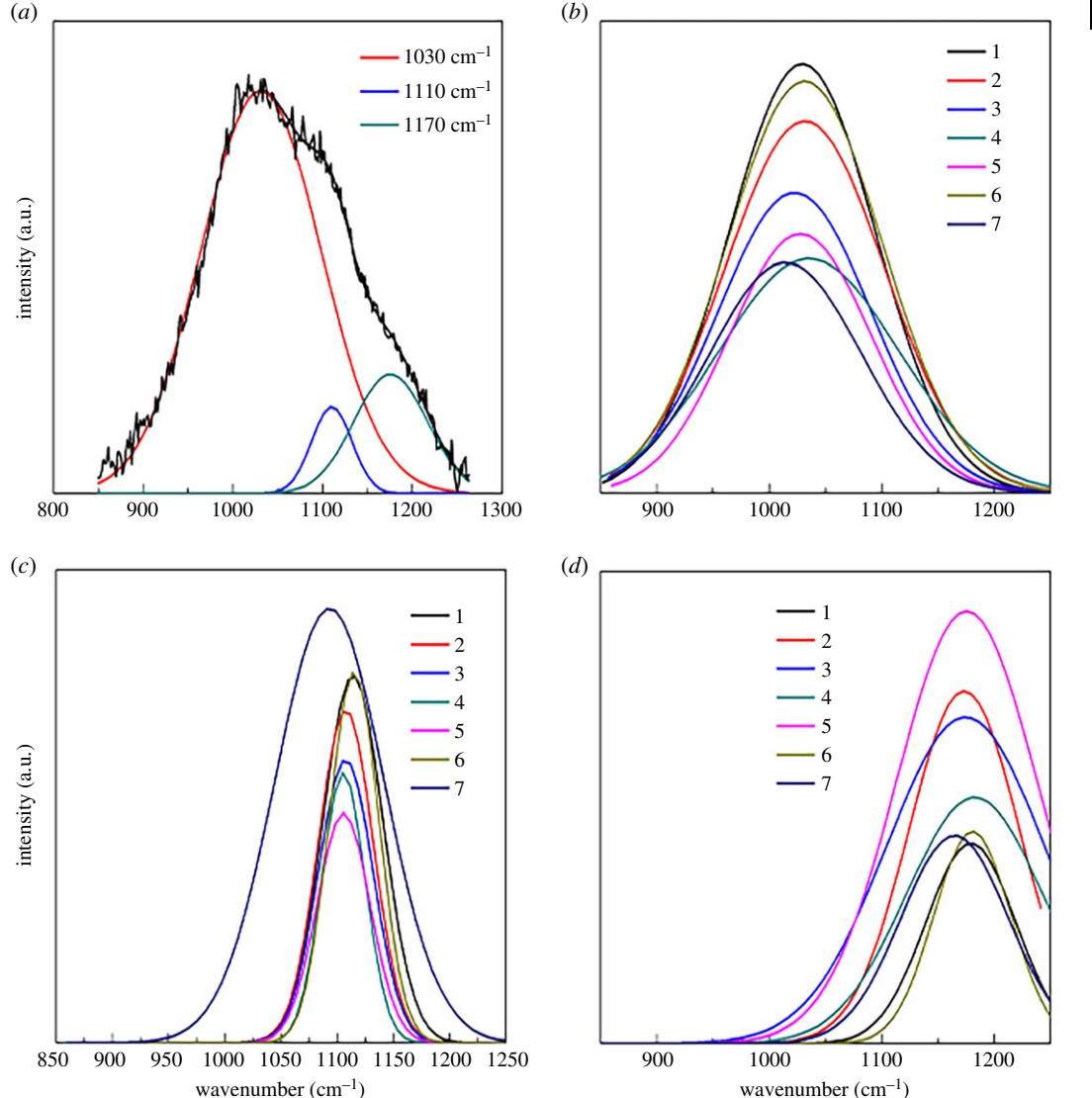

**Figure 2.** Gaussian fit of Raman peaks: (a) 850 ∼ 1250 cm$^{-1}$; (b) around 1030 cm$^{-1}$; (c) around 1100 cm$^{-1}$; and (d) around 1170 cm$^{-1}$.

As discussed above, $Q^3$ accounted for the most for the peak at around 1030 cm$^{-1}$, the weakening of the peaks indicated that $Q^3$ decreased with the substitution for $SiO_2$ for samples 2, 3, 4, 5 and 7. So peaks located at around 1110 cm$^{-1}$ relating to vibration of Si–$O_{nb}$ in the $Q^3$ unit also presented a lower frequency shift and lower intensity tendency except sample 7 with $Zr^{4+}$ as shown in figure 2c.

Meanwhile, as shown in figure 2d, peaks located at around 1170 cm$^{-1}$ strengthened with higher intensity but lower frequency shift for samples 2, 3, 4, 5 and 7, indicating the increase of the $Q^4$ unit, but with more disorder. This can be explained as follows.

Firstly, sample 6 shows the same $SiO_2$, $Al_2O_3$ and $B_2O_3$ contents as sample 1, so the peaks kept almost the same as sample 1.

Secondly, compared with reference sample 1, samples 2 and 4 show a higher ratio of Al/Si. With the replacement of $Si^{4+}$ by $Al^{3+}$, some non-bridging oxygens will coordinate $Al^{3+}$ to form an [$AlO_4$] tetrahedron to connect with the [$SiO_4$] tetrahedron in the glass network, and these non-bridging oxygens are the very ones from the $Q^3$ structure unit. So with the increase of Al/Si, the $Q^3$ unit decreased and the $Q^4$ unit increased. In the same way, for samples 3 and 5, some $Si^{4+}$ were replaced by $Ge^{4+}$ or $P^{5+}$, accordingly. The $Q^3$ unit decreased with the decrease of $SiO_2$ and can be connected with the [$GeO_4$] or [$PO_4$] tetrahedron to form the $Q^4$ unit. So the strengthening of $Q^4$ vibration probably came from Si–$O_b$–Al caused by the formation of an [$AlO_4$] tetrahedron for samples 2 and 4, and Si–$O_b$–Ge, Si–$O_b$–P for samples 3 and 5. This is why the Si–$O_{nb}$ vibration at around 1110 cm$^{-1}$ weakened, while the Si–$O_b$ vibration at 1170 cm$^{-1}$ strengthened for samples 2, 3, 4 and 5. Basically, with the higher ratio of Al/Si, Si–$O_b$–Si and Si–$O_b$–B decreased, while Si–$O_b$–Al increased, along with

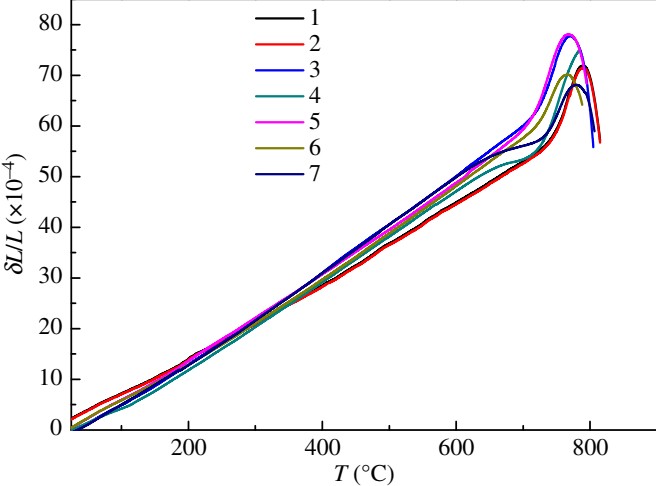

**Figure 3.** Thermal expansion curves of TFT-LCD substrate glass.

**Table 3.** Coefficient of thermal expansion of TFT-LCD substrate glass.

| sample | coefficient of thermal expansion (E-6 °C$^{-1}$) (20–300°C) | thermal expansion softening temperature (°C) |
|---|---|---|
| 1 | 3.60 | 791 |
| 2 | 3.63 | 794 |
| 3 | 3.68 | 771 |
| 4 | 3.64 | 784 |
| 5 | 3.70 | 767 |
| 6 | 3.57 | 776 |
| 7 | 3.52 | 782 |

the newcomer Si–$O_b$–Ge or Si–$O_b$–P, bringing more distortion into the glass network compared with sample 1, causing the increase of the disorder degree of the network and lower frequency shift of the peaks.

Finally, sample 7 with $Zr^{4+}$ showed the more obvious shift of peaks at all three frequencies than other samples. On the one hand, sample 7 showed the lowest $Y$ value as shown in table 2, on the other hand, $Zr^{4+}$ shows high electric field and coordinate ability, which can enter the network in the form of six coordinates in the $[ZrO_6]$ octahedron and affect the network more significantly than other samples [20]. As a result of six coordination numbers for $Zr^{4+}$, free oxygen ions and the $[SiO_4]$ tetrahedron will reduce more than other samples. So peaks in this region for sample 7 shifted to lower frequency than others. This suggested that the non-bridging oxygen vibration accounted for a larger per cent for sample 7 than other samples, in accordance with the $X$ value as shown in table 2.

It can be derived and inferred from above, the low frequency bending vibration of Si–$O_b$ in Si–$O_b$–Si bonds dominated in alkali-free glass, and $B^{3+}$ was mainly found in the form of a $[BO_3]$ triangle owing to its inferiority to $Al^{3+}$ coordinated by oxygen ions and fewer alkali-earth ions. As for the $[SiO_4]$ tetrahedron, $Q^3$ and $Q^4$ units accounted for the most in four kinds of $[SiO_4]$ units. With the substitution of oxides for $SiO_2$, $Q^3$ decreased and $Q^4$ increased, but the disorder degree of the network increased, embodied in the lower frequency shift of the Raman peak. Sample 7 with $Zr^{4+}$ showed more peak shift than others, mainly owing to the high electric field of $Zr^{4+}$ that can significantly affect the network of glass.

## 3.2. Physico-chemical properties of glass

### 3.2.1. Thermal properties

The thermal expansion properties of samples are shown in table 3 and in figure 3, including the linear expansion coefficients and the thermal expansion softening temperature. Thermal expansion is dominated by the intensity of chemical bonds between oxygen ions and cations expressed as $z/r^2$,

**Table 4.** Mechanical properties of TFT-LCD substrate glass.

| sample | density (g cm$^{-3}$) | elastic modulus (GPa) | shear modulus (GPa) | Poisson's ratio |
|--------|----------------------|-----------------------|---------------------|-----------------|
| 1 | 2.525 | 76.5 | 31.2 | 0.23 |
| 2 | 2.523 | 76.1 | 30.9 | 0.23 |
| 3 | 2.545 | 75.9 | 30.7 | 0.24 |
| 4 | 2.524 | 75.6 | 30.3 | 0.25 |
| 5 | 2.522 | 74.6 | 30.3 | 0.23 |
| 6 | 2.527 | 76.8 | 31.6 | 0.22 |
| 7 | 2.540 | 79.6 | 32.0 | 0.24 |

which are directly proportional to cation charge and inversely proportional to square of distance of them [21]. As a result of the similar structure judged by the Raman spectra above, the linear expansion coefficients for samples changed a little between 3.52 and 3.70 E-6°C$^{-1}$ from 20°C to 300°C.

Chemical bond Ge–O is close to Si–O, but the radius of Ge$^{4+}$ is larger than Si$^{4+}$, leading to a weaker force between Ge$^{4+}$ and O$^{2-}$. So the thermal expansion is bigger for sample 3. In the same way, sample 4 showed a higher coefficient of thermal expansion owing to the weaker force between B$^{3+}$ and O$^{2-}$ than Si$^{4+}$ and O$^{2-}$, along with the increasing [BO$_3$] triangles and decreasing [SiO$_4$] tetrahedrons which could lower the compactness of the network. Sample 5 exhibited the highest coefficient arising from the existence of P=O double bond, which is an asymmetry centre that could bring unsteadiness to the structure. Meanwhile, sample 6 also showed a relatively lower coefficient than sample 1. Compared with the reference sample 1, sample 6 contains a smaller amount of SrO and larger amount of CaO. The radius of Sr$^{2+}$ is larger than Ca$^{2+}$. The intensity of chemical bond between the alkali-earth and oxygen ion will increase with the decrease of SrO and increase of CaO, which could make the network become more compacted and the coefficient of thermal expansion decrease.

Sample 7 showed the lowest coefficient, probably owing to the existence of the high electric field ions of Zr$^{4+}$. The modified cation Zr$^{4+}$ can be coordinated with six oxygen ions to form a [ZrO$_6$] octahedron to connect with the [SiO$_4$] tetrahedron [20], which can increase the compactness of the network and weaken the process of thermal expansion.

The thermal expansion softening temperature were between 767°C and 791°C for seven samples. They also showed a good agreement with the tightness of the glass network as discussed above.

### 3.2.2. Mechanical properties

The mechanical properties of glass are shown in table 4, including density, elastic modulus, shear modulus and Poisson's ratio.

As a result of closer architecture of the network discussed above, density of samples investigated in this paper are mainly dominated by the specific gravity of oxides. Firstly, more Al$_2$O$_3$ will bring more of the [AlO$_4$] tetrahedron with larger volume than the [SiO$_4$] tetrahedron, making the network loose, but the weight of Al$_2$O$_3$ is bigger than SiO$_2$. So the competing effect of them does not make the density change too much comparing sample 2 and 1 for 1% mole fraction substitution. Similarly, samples 3 and 7 showed larger density than other samples, mainly arising from the more weighted GeO$_2$ and ZrO$_2$ than SiO$_2$. A 1% mole fraction substitution of GeO$_2$ for SiO$_2$, as well as ZrO$_2$ for SiO$_2$ could make the density increase by about 0.8%. Furthermore, the accumulation effect of Zr$^{4+}$ and the connection of the [ZrO$_6$] octahedron and the [SiO$_4$] tetrahedron could make the network more compacted, which could also increase the density. Samples 4 and 5 also showed the density near 2.520 g cm$^{-3}$, owing to the closer weight between B$_2$O$_3$, P$_2$O$_5$ and SiO$_2$. Meanwhile, sample 6 showed higher density than sample 1, mainly owing to heavier Sr$^{2+}$ than Ca$^{2+}$, as well as a more compacted structure as discussed above.

The elastic modulus and shear modulus are dominated by the intensity of chemical bonds between oxygen ions and cations, which are directly proportional to cation charge and inversely proportional to square of distance of them. Sample 7 showed the largest value owing to the existence of the high electric field ions of Zr$^{4+}$ as discussed above. Sample 6 exhibited a little higher modulus than sample 1, owing to the larger radius for Sr$^{2+}$ than Ca$^{2+}$ for the same reason as above. The weaker force between O$^{2-}$ and Ge$^{4+}$ than Si$^{4+}$ lead to the decrease of modulus with the substitution of GeO$_2$ for SiO$_2$ compared with

**Table 5.** Optical properties of TFT-LCD substrate glass.

| sample | transmittance (%) (400–800 nm) | refractive index |
|---|---|---|
| 1 | 90.14 | 1.522 |
| 2 | 90.09 | 1.524 |
| 3 | 90.05 | 1.521 |
| 4 | 90.76 | 1.521 |
| 5 | 90.20 | 1.521 |
| 6 | 90.79 | 1.523 |
| 7 | 89.98 | 1.526 |

**Table 6.** Electrical properties of TFT-LCD substrate glass.

| sample | logarithm volume resistivity ($\Omega$ cm) | |
|---|---|---|
| | 25°C | 250°C |
| 1 | 15.9 | 13.1 |
| 2 | 15.8 | 13.2 |
| 3 | 15.7 | 12.6 |
| 4 | 15.5 | 13.0 |
| 5 | 15.1 | 12.8 |
| 6 | 15.5 | 12.8 |
| 7 | 15.3 | 12.8 |

samples 2 and 1. For sample 6, the modulus reduced compared with sample 1, probably arising from the existence of the asymmetric centre of a P=O double bond.

### 3.2.3. Optical properties

Table 5 lists the optical properties of the seven samples, including the average transmittance in visible light and the refractive index. All seven samples were cut and ground to the same thickness of 2 mm. As shown in table 5, the average transmittance of seven samples were very close to each other with the values of 90–91%, as well as the refractive index at around 1.520–1.526, showing very small differences with each other. From the analysis of the Raman spectra above, the glass network of all samples with the slight adjustment of oxides did not show too much difference. So the optical properties which were mainly dominated by the network and compositions exhibited similar performances accordingly.

Meanwhile, sample 7 showed a little smaller transmittance and a little larger refractive index than other samples. The extranuclear electrons of $Zr^{4+}$ were easily polarized owing to the larger radius of $Zr^{4+}$, and the refractive index of glass relates to the polarity of electrons. The increasing refractive index also affects the transmittance [22]. As a result of small amount of substitution of $ZrO_2$ for $SiO_2$ for sample 7, the effect of $Zr^{4+}$ was not too significant.

### 3.2.4. Electrical properties

Table 6 shows the logarithm volume resistivity of samples measured at 25°C and 250°C, respectively. TFT-LCD substrate glass is a kind of alkali-free glass and the alkali-earth ions are the primary conductive elements, which show a lower conductivity than alkaline ions. This is why the volume resistivity of TFT-LCD substrate glass is larger than conventional soda lime glass. At a temperature of 25°C, the logarithm volume resistivities the TFT-LCD substrate glass is between 15.1 $\Omega$ cm and 15.9 $\Omega$ cm. When the samples were heated, the energy of alkali-earth ions increased, and the number of ions with high enough energy in the glass network that could migrate in the electric field also

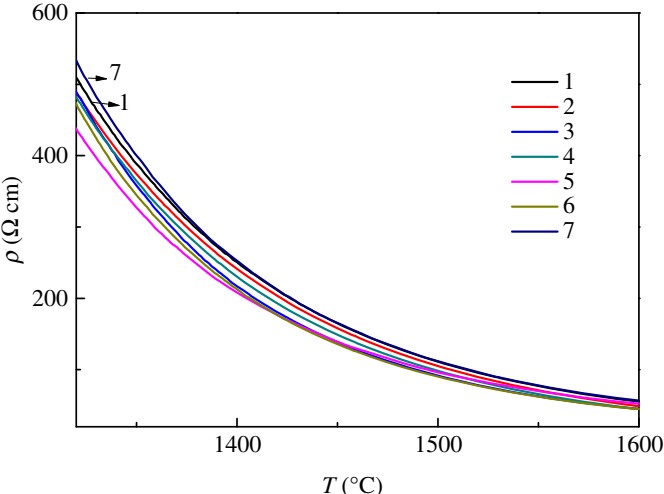

**Figure 4.** Molten resistivity curves of TFT-LCD substrate glass.

increased, causing the decrease of the resistivity to around 13 $\Omega$ cm at 250°C shown in table 5, which declined about 100 times compared with that at 25°C.

### 3.2.5. High temperature resistivity

The curves of high temperature resistivity for the seven samples are shown in figure 4. The resistivities decreased with the temperature ranging from 1300°C to 1600°C. When the temperature was not high enough to destroy the network of glass, for example, at 1300–1400°C, alkali-earth ions will move in the gap of the network to conduct.

Sample 2 showed smaller resistivity than sample 1. As a result of different volumes of the [AlO$_4$] and [SiO$_4$] tetrahedrons, the connection between them will bring more gaps. Alkali-earth ions will move easier with increasing Al$_2$O$_3$ and decreasing SiO$_2$ and make resistivity reduced.

As for the comparison of samples 3 and 1, the resistivity became lower by the substitution of GeO$_2$ for SiO$_2$. Because the volume of [GeO$_4$] is larger than the [SiO$_4$] tetrahedron, the network of glass expands to generate a more relaxed structure, which benefited from the movement for alkali-earth ions and reduced the resistivity for sample 3.

Sample 4 exhibited a lower resistivity than sample 1. The amount of B$_2$O$_3$ is higher for sample 4 than sample 1, and the number of the [BO$_3$] triangle is larger, correspondingly. The network with higher [BO$_3$] triangles was more loose, resulting to smaller resistivity.

The one with the lowest resistivity was sample 5 owing to the phosphorus and oxygen double bond P=O in the network brought about by the substitution of P$_2$O$_5$ and SiO$_2$, which introduces the instability in the network. Alkali-earth ions can move smoothly in the field of a relaxed network, leading to the decrease of resistivity.

Sample 6 exhibited a smaller resistivity compared with sample 1, probably owing to the different mobilities for Ca$^{2+}$ and Sr$^{2+}$. The radius of Sr$^{2+}$ is bigger than that of Ca$^{2+}$. This will make it more difficult to move in the network for Sr$^{2+}$ than for Ca$^{2+}$, and the mobility of Sr$^{2+}$ is lower than Ca$^{2+}$. So the resistivity of sample 6 was smaller owing to its more CaO and less SrO content compared with sample 1.

Sample 7 showed a higher resistivity than sample 1, mainly owing to the existence of Zr$^{4+}$. The connection of the [ZrO$_6$] octahedron and the [SiO$_4$] tetrahedron may bring some barriers for the movement of alkali-earth ions and cause the increase of resistivity.

Finally, with the increase of temperature, the glass network vanished for all seven samples, and the mechanism of conduction was the free movement of alkali-earth ions in molten glass liquid, leading to the same value in the final molten state for about 50 $\Omega$·cm for all seven samples whose number of alkali-earth ions were equal.

### 3.2.6. Viscosity-temperature characteristics

Figure 5 shows the curves of high temperature viscosity of the seven samples in the range of 1100–1700°C, wherein data ranging from 1200°C to 1550°C were the measured values, corresponding to viscosity of $10^{1.5}$–$10^{3.2}$ Pa s, and data beyond that were fitted by the Fulcher equation [23,24]. As a matter of fact, the

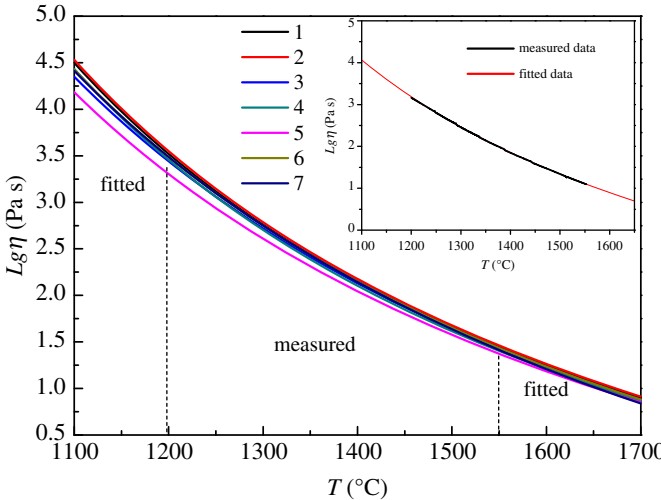

**Figure 5.** High temperature viscosity of TFT-LCD substrate glass.

**Table 7.** Reference points of viscosity-temperature characteristics for the float process. ($T_m$: 10 Pa s; $T_f$: $10^2$ Pa s; $T_w$: $10^3$ Pa s; polishing area: $10^{2.7}$–$10^{3.2}$ Pa s; slow cooling area: $10^{3.2}$–$10^{4.25}$ Pa s.)

| sample | $T_m$ (°C) | $T_f$ (°C) | $T_W$ (°C) | polishing area (°C) | slow cooling area (°C) |
|---|---|---|---|---|---|
| 1 | 1669 | 1429 | 1266 | 1309–1239 | 1239–1123 |
| 2 | 1674 | 1432 | 1272 | 1313–1241 | 1241–1126 |
| 3 | 1661 | 1427 | 1252 | 1304–1235 | 1235–1110 |
| 4 | 1658 | 1422 | 1257 | 1300–1231 | 1231–1117 |
| 5 | 1651 | 1412 | 1241 | 1286–1211 | 1211–1094 |
| 6 | 1665 | 1428 | 1262 | 1308–1237 | 1237–1116 |
| 7 | 1677 | 1433 | 1271 | 1314–1243 | 1243–1125 |

difference between measured data and fitted data was less than 1°C. So in figure 5, only the smooth curves of fitted data were shown. Table 7 shows some reference points of viscosity-temperature characteristics for float glass, including melting point of glass ($T_m$), forming point of glass ($T_f$), working point of glass ($T_w$) and the range for polishing area and slow cooling area.

As shown in figure 5, all samples showed very closer curves of viscosity as a result of the similar network as discussed above, but the short-range disorder caused by the substitution of oxides may bring some differences in the properties of viscosity.

Firstly, $Al^{3+}$ entered the network in the form of an $[AlO_4]$ tetrahedron as a result of $RO/Al_2O_3 > 1$. The small amount substitution of $Al_2O_3$ for $SiO_2$ will increase the viscosity. By comparison of sample 2 and sample 1, melting point increased for 5°C and working point increased for 6°C with the substitution of 1% mole fraction.

Secondly, $GeO_2$ is easier to become molten than $SiO_2$, and the substitution of $GeO_2$ for $SiO_2$ will bring the short-range disorder as a result of the different volumes of $[GeO_4]$ and $[SiO_4]$ tetrahedrons in the glass network, making the network loose and the reference points of viscosity-temperature characteristics decrease. By comparison of sample 3 and sample 1, melting point decreased for 8°C and working point decreased for 14°C with the substitution of 1% mole fraction.

Sample 4 was acquired by the substitution of $B_2O_3$ for $SiO_2$ with 1% mole fraction. As a result of $(RO-Al_2O_3)/B_2O_3 < 1$, with the increase of $B_2O_3$ and decrease of $SiO_2$, the $[SiO_4]$ tetrahedron decreased and the $[BO_3]$ triangle increased, increasing the disorder and lowering the compactness of the network, and thus making the viscosity of glass liquid reduced. By comparison of sample 4 and sample 1, with the increase of $B_2O_3$ and decrease of $SiO_2$ for 1% mole fraction, melting point and working point decreased for 11°C and 9°C, respectively.

For sample 5 and sample 1 with the substitution of $P_2O_5$ for $SiO_2$, the reference points of viscosity-temperature characteristics decreased more than the other samples. $P_2O_5$ can lower the melting point

**Table 8.** Strain points, annealing points and softening points of TFT-LCD substrate glass. ($T_{st}$: strain point; $T_a$: annealing point; $T_s$: softening point.)

| sample | $T_{st}$ (°C) | $T_a$ (°C) | $T_s$ (°C) |
|---|---|---|---|
| 1 | 676.2 | 724.3 | 962.3 |
| 2 | 678.1 | 725.9 | 963.8 |
| 3 | 671.1 | 722.7 | 952.5 |
| 4 | 670.3 | 720.1 | 947.6 |
| 5 | 667.1 | 716.2 | 946.2 |
| 6 | 669.5 | 719.3 | 954.8 |
| 7 | 680.3 | 727.2 | 969.3 |

probably owing to its small viscous activation energy, meanwhile, $P^{5+}$ can form a [$PO_4$] tetrahedron and enter the network with the P=O double bond, which is an asymmetry centre and will make the structure relaxed and thus lower the viscosity of the glass liquid. By comparison of sample 5 and sample 1, melting point and working point decreased for 18°C and 25°C, respectively, exhibiting the most decrease for all samples.

Although the network formers and intermediate formers are the same for samples 6 and 1, the substitution of alkali-earth ions could also bring some difference in viscosity. The radius of $Sr^{2+}$ is bigger than $Ca^{2+}$, the intensity of the chemical bond became stronger with increasing CaO and decreasing SrO, thus the ability of depolymerizing the big tetrahedron group can increase and capturing the oxygen ions distributing them around, thus the network becomes more relaxed and the viscosity decreases. So the viscosity of sample 6 was smaller than sample 1. Melting point and work point decreased for 4°C and 4°C, respectively. This suggests that the effect by the substitution of modified cations showed less effect than network formers.

Finally, comparing sample 7 and sample 1 with the substitution of $ZrO_2$ for $SiO_2$, the connection of the [$ZrO_6$] octahedron and the [$SiO_4$] tetrahedron made the network more compacted as discussed above. The result showed that with the substitution of 1% mole fraction, melting point and working point increased for 8°C and 5°C, respectively

Table 8 shows the strain points, annealing points and softening points for all seven samples. The change with the adjustment of oxides was basically consistent with the results discussed above. They were consistent with the compactness of the network.

## 4. Conclusion

By the slight adjustment of oxides of the TFT-LCD substrate glass, including equal mole fraction substitution of $Al_2O_3$, $GeO_2$, $B_2O_3$, $P_2O_5$ and $ZrO_2$ for $SiO_2$, as well as the substitution of CaO for SrO with the total contents unchanged, the structure and physico-chemical properties of glass was investigated. The results were as follows.

(i) The architecture of the glass network did not show too much difference with the substitution of 1% mole fraction for network formers and modified cations. The low frequency bending vibration of Si–$O_b$ in Si–$O_b$–Si bonds dominated in alkali-free glass, and $B^{3+}$ was mainly found in the form of a [$BO_3$] triangle owing to its inferiority to $Al^{3+}$ coordinated by oxygen ions and fewer alkali-earth ions. As for the [$SiO_4$] tetrahedron, $Q^3$ and $Q^4$ units accounted for the most in four kinds of [$SiO_4$] units. With the substitution of oxides for $SiO_2$, $Q^3$ decreased and $Q^4$ increased, but the disorder degree of network increased, embodied in the lower frequency shift of the Raman peak. Sample 7 with $Zr^{4+}$ showed more peak shift than others, mainly owing to the high electric of $Zr^{4+}$ that can significantly affect the glass network.

(ii) The stability and entirety of the glass network deteriorated by substitution of $GeO_2$, $B_2O_3$, $P_2O_5$ for $SiO_2$, and some physiochemical property indexes got worse, such as the decrease of elastic modulus and shear modulus for the mechanical properties, the increase of the thermal expansion coefficient for the thermal properties, and the decrease of the reference points of viscosity-temperature characteristics. Accordingly, the process for float glass became easier by the substitution, which presented a contrary tendency. For example, the melting point of the glass and the resistivity of glass liquid both decreased, which would contribute to the melting process of glass.

(iii) For the substitution of $Al_2O_3$ and $ZrO_2$ for $SiO_2$, the stability of the network did not show significant deterioration, probably owing to the $[AlO_4]$ tetrahedron and the $[ZrO_6]$ octahedron. They can enter the network to connect with the $[SiO_4]$ tetrahedron and the compactness could be strengthened and the property indexes correspondingly became better, such as the increase of elastic modulus and shear modulus for the mechanical properties, the decrease of the thermal expansion coefficient for the thermal properties, and the increase of the viscosity-temperature characteristics. Meanwhile, the process for float glass became harder by the substitution. For example, the melting point of glass and the resistivity of glass liquid both increased, bringing more difficulties to the melting process. As for the 1% mole fraction substitution of oxides investigated above, all the values exhibited variations in a reasonable range.

Data accessibility. Data are available from the Dryad Digital Repository: https://doi.org/10.5061/dryad.9w0vt4b9z [25].

Authors' contributions. J.C. designed the study, performed the laboratory experiments and wrote the manuscript. L.S. and Z.Z. assisted in analysing experimental data. X.C. assisted in editing the manuscript for important intellectual content, and gave the final approval for publication. P.W. was responsible for the revision of the article language. L.M. was responsible for the arrangement of the entire project.

Competing interests. We declare we have no competing interests.

Funding. We gratefully acknowledge the support of this research by the National Key Research and Development Program of China (grant no. 2016YFB0303700).

Acknowledgements. We are grateful to Mr Tianhe Wang for helping us to complete the experiment.

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
