## [Reviewer comments · Royal Society Open Science]

Review History

RSOS-191425.R0 (Original submission)

Review form: Reviewer 1

Is the manuscript scientifically sound in its present form?

Yes

Are the interpretations and conclusions justified by the results?

Yes

Is the language acceptable?

No

Do you have any ethical concerns with this paper?

No

Have you any concerns about statistical analyses in this paper?

Yes

Recommendation?

Accept with minor revision (please list in comments)

Comments to the Author(s)

The authors prepared and studied the structure of alkali-free glasses ($\text{RO-B}_2\text{O}_3\text{-Al}_2\text{O}_3\text{-SiO}_2$) as well as described their thermal properties, mechanical properties, electrical resistivity, viscosity and glass network structure over a small composition changing. There is no doubt that the motivation of this work is interesting, and this work adds some insights in this research area. I suggest that the authors take into account the detailed comments below and resubmit a corrected version.

- Please delete Fig.2. Usually, the intensity of Raman bands makes no sense, especially for such small intensity change.
- Some spelling mistakes, for example: line 13, page 7, "obviouly"; line13, page16, "fracton", table 6, "No". Please check through the paper.
- Please add the temperature range for thermal expansion of glasses. (Table 3)
- The viscosity of glass at high temperature could be well described by Arrhenius equation. The authors used Fulcher equation to fit the viscosity data, and the measured data and fitted data did not be distinguished (Fig.6). The smooth curves look like all from fitting results.

Review form: Reviewer 2

Is the manuscript scientifically sound in its present form?

Yes

Are the interpretations and conclusions justified by the results?

Yes

Is the language acceptable?

Yes

Do you have any ethical concerns with this paper?

No

Have you any concerns about statistical analyses in this paper?

No

Recommendation?

Accept with minor revision (please list in comments)

Comments to the Author(s)

This manuscript investigates the influence of different additive oxides on the structural (by Raman) and physicochemical properties (thermal, mechanical, optical, electrical, rheological) of TFT-LCD substrate glasses. Overall it is a careful study and the manuscript is clear and precise.

My questions and comments:

P3 "The samples with megascopic homogeneity and invisible bubbles were selected..."
It sounds weird. I think the authors meant "samples without visible bubbles".

P3 2.3 Measurements

Both manufacturer and model should be included for each of the instruments.

P9 "The thermal expansion properties of samples are shown in table 3 and in figure 4, including the coefficient of thermal expansion..."

Please specify they are linear or volume expansion coefficients.

P6 "cmc values decrease at low content of surfactant..."

This sentence is confusing. Cmc value is the surfactant concentration above which micelles form. This sentence is like saying "The boiling point of water decrease at low temperature."

P17 Acknowledgements. "I am grateful to Mr Tianhe Wang for helping me to complete the experiment."

"We" should be used instead of "I".

Decision letter (RSOS-191425.R0)

04-Oct-2019

Dear Mr Cui:

Title: Influence of the Slight Adjustment of Oxides on the Structural and Physicochemical Properties of TFT-LCD substrate glasses
Manuscript ID: RSOS-191425

Thank you for submitting the above manuscript to Royal Society Open Science. On behalf of the Editors and the Royal Society of Chemistry, I am pleased to inform you that your manuscript will be accepted for publication in Royal Society Open Science subject to minor revision in accordance with the referee suggestions. Please find the reviewers' comments at the end of this email.

The reviewers and handling editors have recommended publication, but also suggest some minor revisions to your manuscript. Therefore, I invite you to respond to the comments and revise your manuscript.

Because the schedule for publication is very tight, it is a condition of publication that you submit the revised version of your manuscript before 13-Oct-2019. Please note that the revision deadline will expire at 00.00am on this date. If you do not think you will be able to meet this date please let me know immediately.

Best wishes,
Dr Laura Smith
Publishing Editor, Journals

On behalf of the Subject Editor Professor Anthony Stace and the Associate Editor Professor Tobias Hertel.

RSC Associate Editor:
Comments to the Author:
(There are no comments.)

RSC Subject Editor:
Comments to the Author:
(There are no comments.)

Reviewer comments to Author:

Reviewer: 1

Comments to the Author(s)

The authors prepared and studied the structure of alkali-free glasses ($\text{RO-B}_2\text{O}_3\text{-Al}_2\text{O}_3\text{-SiO}_2$) as well as described their thermal properties, mechanical properties, electrical resistivity, viscosity and glass network structure over a small composition changing. There is no doubt that the motivation of this work is interesting, and this work adds some insights in this research area. I suggest that the authors take into account the detailed comments below and resubmit a corrected version.

- Please delete Fig.2. Usually, the intensity of Raman bands makes no sense, especially for such small intensity change.
- Some spelling mistakes, for example: line 13, page 7, "obviouly"; line13, page16, "fracron", table 6, "No". Please check through the paper.
- Please add the temperature range for thermal expansion of glasses. (Table 3)
- The viscosity of glass at high temperature could be well described by Arrhenius equation. The authors used Fulcher equation to fit the viscosity data, and the measured data and fitted data did not be distinguished (Fig.6). The smooth curves look like all from fitting results.

Reviewer: 2

Comments to the Author(s)

This manuscript investigates the influence of different additive oxides on the structural (by Raman) and physicochemical properties (thermal, mechanical, optical, electrical, rheological) of TFT-LCD substrate glasses. Overall it is a careful study and the manuscript is clear and precise.

My questions and comments:

P3 "The samples with megascopic homogeneity and invisible bubbles were selected..."
It sounds weird. I think the authors meant "samples without visible bubbles".

P3 2.3 Measurements

Both manufacturer and model should be included for each of the instruments.

P9 "The thermal expansion properties of samples are shown in table 3 and in figure 4, including the coefficient of thermal expansion..."

Please specify they are linear or volume expansion coefficients.

P6 "cmc values decrease at low content of surfactant..."

This sentence is confusing. Cmc value is the surfactant concentration above which micelles form. This sentence is like saying "The boiling point of water decrease at low temperature."

P17 Acknowledgements. "I am grateful to Mr Tianhe Wang for helping me to complete the experiment."

"We" should be used instead of "I".

Author's Response to Decision Letter for (RSOS-191425.R0)

See Appendix A.

Decision letter (RSOS-191425.R1)

19-Nov-2019

Dear Mr Cui:

Title: Influence of the Slight Adjustment of Oxides on the Structural and Physicochemical Properties of TFT-LCD substrate glasses
Manuscript ID: RSOS-191425.R1

It is a pleasure to accept your manuscript in its current form for publication in Royal Society Open Science. The chemistry content of Royal Society Open Science is published in collaboration with the Royal Society of Chemistry.

On behalf of the Subject Editor Professor Anthony Stace and the Associate Editor Professor Tobias Hertel.

RSC Associate Editor
Comments to the Author:
(There are no comments.)

Reviewer(s)' Comments to Author:

Appendix A

Influence of the Slight Adjustment of Oxides on the Structural and Physicochemical Properties of TFT-LCD substrate glasses

List of Responses

Dear Editors and Reviewers:

Thank you for your letter and for the reviewers' comments concerning our manuscript entitled "Influence of the Slight Adjustment of Oxides on the Structural and Physicochemical Properties of TFT-LCD substrate glasses" (ID: RSOS-191425).

Those comments are all valuable and very helpful for revising and improving our paper, as well as the important guiding significance to our researchs. We have studied comments carefully and have made correction. Revised portion are marked in red in the paper. The main corrections in the paper and the responds to the reviewer's comments are as following:

Responds to the reviewer's comments:

Reviewer #1:

1. "Please delete Fig.2. Usually, the intensity of Raman bands makes no sense, especially for such small intensity change".

Response: As Reviewer suggested that Raman bands makes no sense for such small intensity change, we have deleted Fig.2 in our revised paper. And Fig.3 to Fig.6 have been renamed as Fig.2 to Fig.5, and texts in the paper have also been kept as consistent with figures accordingly.

2. "Some spelling mistakes, for example: line 13, page 7, "obviouly"; line13, page16, "fracton", table 6, "No". Please check through the paper".

Response: We are very sorry for our spelling mistakes and we have checked through the paper and made correction as marked in red according to the Reviewer's suggestions.

3. "Please add the temperature range for thermal expansion of glasses. (Table 3) "

Response: Thanks a lot for the Reviewer's suggestion and the temperature range has been added in the revised paper for 20~300°C.

4. "The viscosity of glass at high temperature could be well described by Arrhenius equation. The authors used Fulcher equation to fit the viscosity data, and the measured data and fitted data did not be distinguished (Fig.6). The smooth curves look like all from fitting results".

Response: Thanks very much for the comments. A new figure showing the difference between the measured data and fitted data was added in Fig.5(as a result of the deletion of Fig.2), and it could be seen that they showed very closer value.As a matter of fact, the measured data and fitted data was less than 1°C.So in the revised paper, it has been specified to fitted curves for Fig.5 for all samples. Meanwhile, the Orton viscometer adopted Fulcher equation to fit the viscosity

data, so in this paper we also choosed the defaulted equation for our research.

Special thanks to you for your valuable comments.

Reviewer #2:

1. P3 “The samples with megascopic homogeneity and invisible bubbles were selected...”It sounds weird. I think the authors meant "samples without visible bubbles".

Response: It is really true as Reviewer suggested that it sounds weird, and we have modified to “samples without visible bubbles ”according to the Reviewer’s suggestion.

2. ”P3 2.3 Measurements Both manufacturer and model should be included for each of the instruments”.

Response: Considering the Reviewer’s suggestion, we have added both manufacturer and model for each of the instruments as marked in red in the revised paper.

3. “P9 “The thermal expansion properties of samples are shown in table 3 and in figure 4, including the coefficient of thermal expansion...”Please specify they are linear or volume expansion coefficients”.

Response: Considering the Reviewer’s suggestion, we have specified the coefficients as the linear expansion coefficients in the paper.

4. “P6 “cmc values decrease at low content of surfactant...”

This sentence is confusing. Cmc value is the surfactant concentration above which micelles form. This sentence is like saying “The boiling point of water decrease at low temperature.”

Response: This part did not belong to our manuscript, I think this may come from another manuscript in the Reviewer, please check it. Thanks a lot.

5. “P17 Acknowledgements. “I am grateful to Mr Tianhe Wang for helping me to complete the experiment.” "We" should be used instead of "I””.

Response: “We are” has been used instead of “I am” according to the Reviewer’s comments.

Special thanks to you for your valuable comments.

We tried our best to improve the manuscript and made some changes in the manuscript. These changes will not influence the content and framework of the paper. And here we did not list the changes but marked in red in revised paper.

We appreciate for Editors/Reviewers’warm work earnestly, and hope that the correction will meet with approval. Once again, thank you very much for your comments and suggestions.

Thank you very much for your attention and consideration.

Sincerely yours,

Corresponding author: Tel: +086-13855223692

Fax: +086-0552-4081011

E-mail address: jdcui04@163.com(JieDong Cui)